# Investigation of Ga_2_O_3_-Based Deep Ultraviolet Photodetectors Using Plasma-Enhanced Atomic Layer Deposition System

**DOI:** 10.3390/s20216159

**Published:** 2020-10-29

**Authors:** Shao-Yu Chu, Meng-Xian Shen, Tsung-Han Yeh, Chia-Hsun Chen, Ching-Ting Lee, Hsin-Ying Lee

**Affiliations:** 1Department of Photonics, National Cheng Kung University, Tainan 701, Taiwan; l78081517@ncku.edu.tw (S.-Y.C.); s230813355@gmail.com (M.-X.S.); ctlee@ee.ncku.edu.tw (C.-T.L.); 2Department of Electrical and Electronic Engineering, Chung Cheng Institute of Technology, National Defense University, Taoyuan 335, Taiwan; cciti09001@ndu.edu.tw; 3Department of Optometry, Chung Hwa University of Medical Technology, Tainan 701, Taiwan; chen1985ch@mail.hwai.edu.tw; 4Department of Electrical Engineering, Yuan Ze University, Taoyuan 320, Taiwan

**Keywords:** detectivity, gallium oxide films, plasma-enhanced atomic layer deposition system, metal-semiconductor-metal ultraviolet C photodetectors, photoresponsivity, X-ray photoelectron spectroscopy spectra

## Abstract

In this work, Ga_2_O_3_ films were deposited on sapphire substrates using a plasma-enhanced atomic layer deposition system with trimethylgallium precursor and oxygen (O_2_) plasma. To improve the quality of Ga_2_O_3_ films, they were annealed in an O_2_ ambient furnace system for 15 min at 700, 800, and 900 °C, respectively. The performance improvement was verified from the measurement results of X-ray diffraction, X-ray photoelectron spectroscopy, and photoluminescence spectroscopy. The optical bandgap energy of the Ga_2_O_3_ films decreased with an increase of annealing temperatures. Metal-semiconductor-metal ultraviolet C photodetectors (MSM UVC-PDs) with various Ga_2_O_3_ active layers were fabricated and studied in this work. The cut-off wavelength of the MSM UVC-PDs with the Ga_2_O_3_ active layers annealed at 800 °C was 250 nm. Compared with the performance of the MSM UVC-PDs with the as-grown Ga_2_O_3_ active layers, the MSM UVC-PDs with the 800 °C-annealed Ga_2_O_3_ active layers under a bias voltage of 5 V exhibited better performances including photoresponsivity of 22.19 A/W, UV/visible rejection ratio of 5.98 × 10^4^, and detectivity of 8.74 × 10^12^ cmHz^1/2^W^−1^.

## 1. Introduction

Due to the ozone hole and thinner ozone layer, a large amount of ultraviolet (UV) light passes through the space to the earth. The UV light seriously damages human skin—UVA (315–400 nm) causes melanin in human skin, UVB (280–315 nm) reduces the skin’s immune system, and UVC (200–280 nm) damages skin cells and causes skin cancer [1,2]. Among them, the deep UV light seriously harms human health. Consequently, UVC photodetectors (UVC-PDs) have received widespread attention. Recently, wide bandgap semiconductors, such as aluminum gallium nitride (AlGaN) [3], magnesium zinc oxide (MgZnO) [4,5], and gallium oxide (Ga_2_O_3_) [6,7], were widely developed and applied to solar-blind UVC-PDs. However, it is difficult to grow high-quality AlGaN epitaxial layers with high Al content due to the quality degradation caused by the more structural defects [8]. Moreover, it is difficult to obtain a high-quality MgZnO film with a bandgap energy larger than 5.0 eV. Since more Zn^2+^ ions are replaced by Mg^2+^ ions in the ZnO lattice as the Mg content increases more than 0.36, the c-axis orientation degrades [9,10]. Because the binary Ga_2_O_3_ has inherent advantages of stable thermal and chemical properties [11], it becomes a promising material for fabricating solar-blind UVC-PDs [12]. Several growth techniques, such as metalorganic chemical vapor deposition (MOCVD) [13,14], pulsed laser deposition (PLD) [15,16], molecular beam epitaxy (MBE) [17,18], magnetron sputtering [19,20,21], and vapor cooling condensation system [22,23], were used to deposit Ga_2_O_3_ films. In this work, a plasma-enhanced atomic layer deposition (PE-ALD) system was used to grow Ga_2_O_3_ films on sapphire substrates. The advantages of the PE-ALD system include low temperature growth, excellent conformality, precise film thickness control, self-limiting growth effect, and large area uniformity [24,25]. Moreover, due to the high reactivity of plasma source on the deposited surface, it provides greater freedom of processing conditions during the deposition process [24]. The material characteristics of the as-grown Ga_2_O_3_ films were measured and analyzed. To further improve the performances of the Ga_2_O_3_ films, the annealing treatment with various temperatures was utilized. The resulting Ga_2_O_3_ films were applied as active layers in metal-semiconductor-metal ultraviolet C photodetectors (MSM UVC-PDs). The performances of the resulting devices with various Ga_2_O_3_ active layers were measured and studied in this work.

## 2. Experimental

The PE-ALD system was used to deposit 80-nm-thick Ga_2_O_3_ films on sapphire substrates. The precursors of trimethylgallium (Ga(CH_3_)_3_, TMGa) and oxygen (O_2_) plasma were used as the Ga and O sources, respectively. One depositing cycle of the Ga_2_O_3_ layer includes four steps. Firstly, to provide Ga source and carry off the residual gas and reactant, TMGa precursor and argon (Ar) gas were sequentially pulsed into the PE-ALD chamber for 0.5 s and 3 s, respectively. After metallization, to provide the O source and carry off the residual gas and reactant, O_2_ plasma and Ar gas were sequentially pulsed into the PE-ALD chamber for 5 s and 3 s, respectively. The chamber pressure and the growth temperature were kept at 0.6 torr and 50 °C, respectively. The growth rate of the Ga_2_O_3_ films was approximately 0.33 Å/cycle. Subsequently, to improve the quality of the Ga_2_O_3_ films, they were annealed in an O_2_ ambient furnace system at various temperatures of 700, 800, and 900 °C for 15 min. The material characteristics of the as-grown and annealing-treated Ga_2_O_3_ films were characterized by X-ray diffraction (XRD), X-ray photoelectron spectroscopy (XPS), photoluminescence (PL) spectroscopy, and ultraviolet-visible (UV-Vis) spectroscopy.

Figure 1 illustrates the schematic configuration of Ga_2_O_3_-based MSM UVC-PDs. To define the active window with a size of 100 × 100 μm^2^ on the Ga_2_O_3_ active layer, the traditional photolithography technique was used. The redundant Ga_2_O_3_ material was etched away using a dilute hydrofluoric acid (HF:H_2_O = 1:25) solution. Interdigitated electrode of Ni/Au (20/100 nm) metals with a spacing of 2 µm and a width of 2 µm was deposited on the patterned Ga_2_O_3_ active layer using an electron-beam evaporator. The current-voltage characteristics of the resulting Ga_2_O_3_-based MSM UVC-PDs were measured using an Agilent 4156C semiconductor parameter analyzer. The photoresponsivity spectra of the prepared MSM UVC-PDs were measured using a monochromator and a deuterium lamp source. The low frequency noise performance was measured using an Agilent 4156C semiconductor parameter analyzer, an HP 35670A dynamic signal analyzer, and a BTA 9812 noise analyzer.

## 3. Results and Discussion

Figure 2 presents the crystallinity of the as-grown Ga_2_O_3_ films and the annealing-treated Ga_2_O_3_ films carried out by XRD with CuKα radiation. Since no diffraction peak was observed in the XRD spectrum of the as-grown Ga_2_O_3_ film, it could be deduced that it was an amorphous structure. There were two diffraction patterns in the XRD spectra of the all annealing-treated Ga_2_O_3_ films. The diffraction peaks at 18.9° and 38.4° corresponded to the ($\bar{2}01$) and ($\bar{4}02$) planes of the β-Ga_2_O_3_, respectively [26]. The intensity of ($\bar{2}01$) and ($\bar{4}02$) peaks increased with the annealing temperature until 800 °C. The phenomenon was attributed to that the suitable temperature annealing treatment could improve the crystallinity of the Ga_2_O_3_ films. However, the intensity of the above-mentioned two peaks decreased with an increase of the annealing temperature to 900 °C. It was deduced that more oxygen vacancies and defects would be induced in the Ga_2_O_3_ films annealed at 900 °C. Consequently, their crystallinity was degraded.

To verify the existence of more oxygen vacancies and defects residing in the Ga_2_O_3_ films annealed at 900 °C, they were measured using an X-ray photoelectron spectroscopy (XPS) and photoluminescence (PL) system. The chemical binding energy of the as-grown Ga_2_O_3_ films and the annealing-treated Ga_2_O_3_ films were analyzed using XPS. Figure 3 shows the O1s core level spectra of the prepared Ga_2_O_3_ films. The O1s peak was composed of two bands located at the binding energy of 530.8 eV and 532.1 eV, which were assigned to the Ga-O bonds and the oxygen vacancy, respectively [27]. The peak ratio of the Ga-O bonds in Ga_2_O_3_ to the oxygen vacancy for the as-grown Ga_2_O_3_ films and the annealing-treated Ga_2_O_3_ films could be derived from the area under the XPS spectra. The peak ratios were 5.92, 7.85, 12.71, and 10.07 for the as-grown Ga_2_O_3_ films and the Ga_2_O_3_ films annealed at 700, 800, and 900 °C, respectively. The peak ratio of the annealing-treated Ga_2_O_3_ films was larger than that of the as-grown Ga_2_O_3_ films, which verified that the oxygen vacancy in the Ga_2_O_3_ films could be improved by annealing treatment in an O_2_ ambience. Besides, since the largest peak ratio was obtained in the Ga_2_O_3_ films annealed at 800 °C, it could be deduced that the lowest oxygen vacancy could be obtained in the Ga_2_O_3_ films annealed at 800 °C.

Figure 4 shows the room-temperature PL spectra of the as-grown Ga_2_O_3_ films and the annealing-treated Ga_2_O_3_ films excited by a He-Cd laser with a wavelength of 325 nm. For all the Ga_2_O_3_ films, the broad band caused by the oxygen vacancies and defects in the visible wavelength region was observed [22,28]. As shown in Figure 4, the PL intensity of the broad band for the Ga_2_O_3_ films decreased with an increase of the annealing temperature until to 800 °C and then increased at a higher annealing temperature of 900 °C. Consequently, it could be deduced that the Ga_2_O_3_ films annealed at 800 °C had the least amount of oxygen vacancies and defects.

The transmission and reflection spectra of the as-grown Ga_2_O_3_ films and the annealing-treated Ga_2_O_3_ films were measured using an UV-Vis spectroscopy. Consequently, from the corresponding transmittance (T) and reflectivity (R), the absorption coefficient (α) was obtained by using the formula of α = ln[(1 − R)^2^/T]/d, where d is the thickness of the films. The optical bandgap energy (E_g_) of the direct bandgap films could be determined using Tauc plot, (αhυ)^2^ = A(hυ − E_g_), where A is a constant and hυ is a photon energy. The curve of (αhν)^2^ versus hυ, as shown in Figure 5, was fitted, and the linear line extended to the photon energy axis to obtain the optical bandgap energy. The optical bandgap energy of 5.07, 5.01, 4.96, and 4.98 eV was obtained for the as-grown Ga_2_O_3_ films and the Ga_2_O_3_ films annealed at 700, 800, and 900 °C, respectively. It was found that the optical bandgap energy of the Ga_2_O_3_ films decreased with an increase of the annealing temperature until to 800 °C and then increased at an annealing temperature of 900 °C. In general, the Ga_2_O_3_ films with the better crystallinity have a lower bandgap energy in comparison with the amorphous Ga_2_O_3_ films [22]. Consequently, the crystallinity of the Ga_2_O_3_ films could be improved by annealing treatment.

Since the wide bandgap Ga_2_O_3_ films deposited using the PE-ALD system were obtained, they were applied to MSM UVC-PDs as active layers. Figure 6 shows the dark current-voltage characteristics of the various Ga_2_O_3_-based MSM UVC-PDs. By illuminating the light with a wavelength of 250 nm and a power intensity of 10.6 µW/cm^2^, the photocurrent of the devices was measured and was shown in Figure 6. Compared with the as-grown Ga_2_O_3_-based MSM UVC-PDs, the dark current and photocurrent of the MSM DUV-PDs with the Ga_2_O_3_ active layer annealed at 800 °C decreased from 43.87 pA to 2.03 pA and increased from 2.77 nA to 27.45 nA, respectively, at bias voltage = 5 V. The improvement in dark current and the photocurrent was attributed to the reduction of oxygen vacancies and defects in the Ga_2_O_3_ film annealed at 800 °C. Furthermore, compared with the MSM UVC-PDs with 800 °C-annealed Ga_2_O_3_ active layer, the larger dark current of 6.01 pA of the devices with the 900 °C-annealed Ga_2_O_3_ active layer at bias voltage = 5 V was attributed to more resided oxygen vacancies and defects. Furthermore, its lower photocurrent of 13.20 nA was attributed to the degraded crystallinity of the 900 °C annealed Ga_2_O_3_ active layer compared with that of the 800 °C annealed Ga_2_O_3_ active layer.

Figure 7 illustrates the photoresponsivity spectra of the various Ga_2_O_3_-based MSM UVC-PDs under a bias voltage of 5 V. The cut-off wavelength and the maximum photoresponsivity of the MSM UVC-PDs with the as-grown Ga_2_O_3_ active layer were about 245 nm and 3.51 A/W, respectively. The cut-off wavelength of the MSM UVC-PDs with the annealing-treated Ga_2_O_3_ active layers was about 250 nm. The corresponded cut-off wavelength of the various Ga_2_O_3_-based MSM UVC-PDs coincided with the associated optical bandgap energy of the used Ga_2_O_3_ active layers. The maximum photoresponsivity of the MSM UVC-PDs with the Ga_2_O_3_ active layers annealed at 700, 800, and 900 °C was 6.17, 22.19, and 12.53 A/W, respectively. Since the Ga_2_O_3_ active layers annealed at 800 °C exhibited the best crystallinity and the lowest oxygen vacancy and defect, its resulting MSM UVC-PD had the best photoresponsivity. The UV (245 nm)/visible (450 nm) rejection ratio of the photoresponsivity (R_rj_) of the MSM UVC-PDs with the as-grown Ga_2_O_3_ active layers was 6.22 × 10^3^. The UV (250 nm)/visible (450 nm) rejection ratio of the photoresponsivity was 1.37 × 10^4^, 5.98 × 10^4^, and 3.09 × 10^4^ for the MSM UVC-PDs with the Ga_2_O_3_ active layers annealed at 700, 800, and 900 °C, respectively. The largest rejection ratio of the MSM UVC-PDs with the 800 °C-annealed Ga_2_O_3_ active layers was attributed to the associated lowest dark current and the largest photoresponsivity resulted from the possession of the best crystallinity and the lowest oxygen vacancy and defect.

Figure 8 shows the frequency-dependent noise power density spectra of the various Ga_2_O_3_-based MSM UVC-PDs under a bias voltage of 5 V. It could be found that the fitting curve of noise power density spectra was consistent with 1/f, where f is frequency. This result demonstrated that the main low-frequency noise was flicker noise. The flicker noise is mainly affected by lattices and surface traps resided in the material. The total noise current (i_n_) could be determined by integrating the noise power density (S_n_(f)):(1)$$i_{n}^{2}=\int_{0}^{\Delta f}S_{n}\left(f\right)\mathrm{df}$$
where △f = 1 kHz is bandwidth. To characterize photodetectors, the important parameters of noise equivalent power (NEP) and detectivity (D*) were respectively calculated as follows:(2)$$\mathrm{NEP}=\frac{\sqrt{i_{n}{}^{2}}}{R}$$
(3)$$D^{*}=\frac{\sqrt{A}\sqrt{△f}}{\mathrm{NEP}}$$
where R is the photoresponsivity and A is the area of active window. The NEP of 5.24 × 10^−13^, 2.24 × 10^−13^, 3.62 × 10^−14^, and 8.79 × 10^−14^ W was obtained for the MSM UVC-PDs with the as-grown Ga_2_O_3_ active layer and the Ga_2_O_3_ active layers annealed at 700, 800, and 900 °C, respectively. The corresponding D* was 6.04 × 10^11^, 1.41 × 10^12^, 8.74 × 10^12^, and 3.60 × 10^12^ cmHz^1/2^W^−1^, respectively. The detectivity of the MSM UVC-PDs with the annealed Ga_2_O_3_ active layers was larger than that of the as-grown Ga_2_O_3_-based MSM UVC-PDs. Furthermore, the detectivity increased with an increase of annealing temperature until 800 °C and then decreased by further increasing annealing temperature to 900 °C. The improvement in detectivity was attributed to that the annealing treatment up to 800 °C could improve the Ga_2_O_3_ crystallinity and could decrease the oxygen vacancies and defects resided in the Ga_2_O_3_ active layer. However, the properties of the 900 °C-annealed Ga_2_O_3_ active layers were degraded. Consequently, due to the obtained best crystallinity and the lowest oxygen vacancies and defects in the 800 °C annealed Ga_2_O_3_ active layers, the resulting MSM UVC-PDs had the largest detectivity.

To further highlight our work, the performance of metal oxide-based MSM PDs with many other similar devices reported previously, as depicted in Table 1. The performance of the MSM UVC-PDs with the 800 °C annealed Ga_2_O_3_ active layers was good.

## 4. Conclusions

In this work, Ga_2_O_3_ films were successfully deposited on sapphire substrate using a PE-ALD system and were applied as active layers in MSM UVC-PDs. The annealing treatment with various temperatures was used to improve the properties of Ga_2_O_3_ films. The XRD experimental results verified that the crystallinity of the Ga_2_O_3_ films could be improved by annealing treatment. Besides, the dependence of oxygen vacancies and defects of the Ga_2_O_3_ films on annealing temperature was verified by XPS and PL experimental results. The best crystallinity and the lowest oxygen vacancy and defect were obtained, when the Ga_2_O_3_ films were annealed in an O_2_ ambience at 800 °C for 15 min. The lowest dark current of 2.03 pA and the largest photoresponsivity of 22.19 A/W were obtained for the MSM UVC-PDs with the 800 °C annealed Ga_2_O_3_ active layers. Compared with the as-grown Ga_2_O_3_-based MSM UVC-PDs, the detectivity of the 800 °C annealed Ga_2_O_3_-based MSM UVC-PDs increased from 6.04 × 10^11^ to 8.74 × 10^12^ cmHz^1/2^W^−1^. Consequently, it is expected that the PE-ALD is a promising system for depositing high-quality Ga_2_O_3_ films, which has high potential in the fabrication of solar-blind deep ultraviolet C photodetectors.

## Figures and Tables

**Figure 1 sensors-20-06159-f001:**
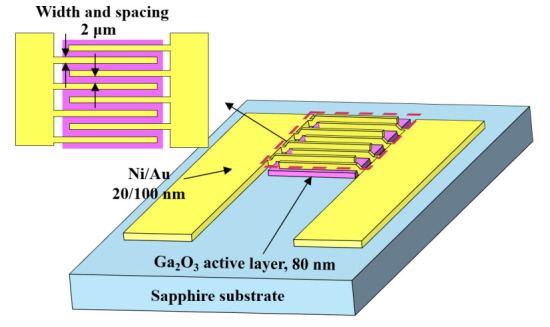
Schematic configuration of metal-semiconductor-metal ultraviolet C photodetectors.

**Figure 2 sensors-20-06159-f002:**
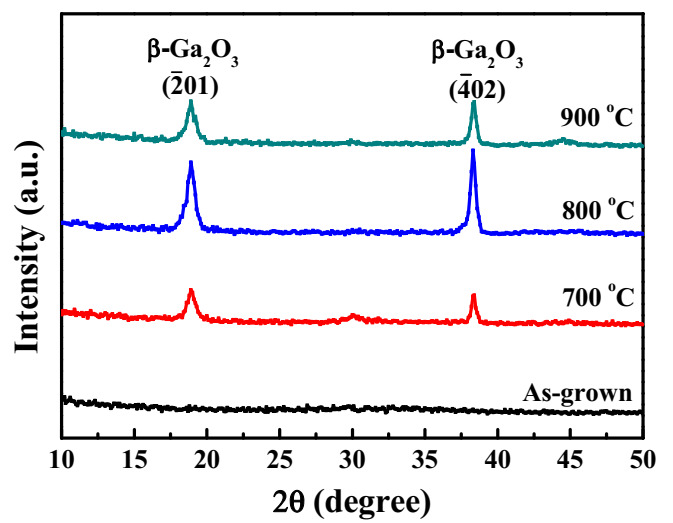
X-ray diffraction (XRD) spectra of Ga_2_O_3_ films annealed at various temperatures.

**Figure 3 sensors-20-06159-f003:**
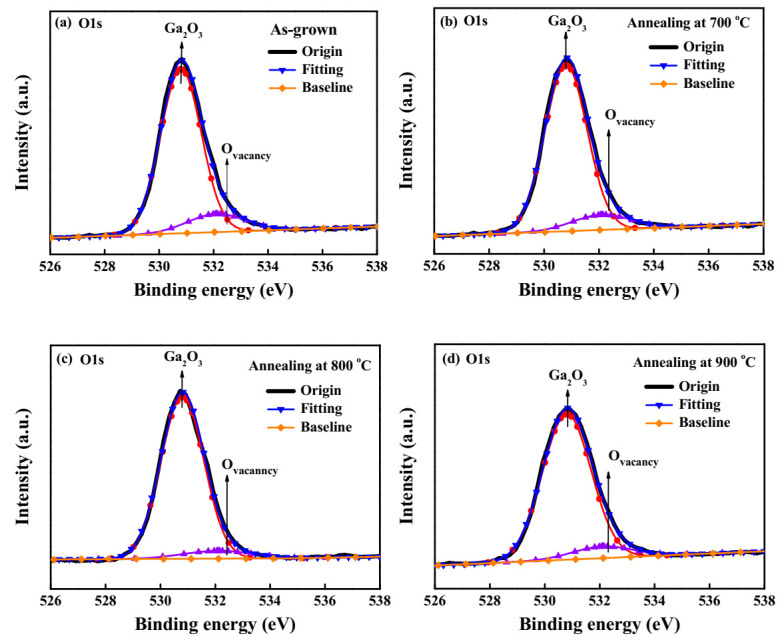
X-ray photoelectron spectroscopy (XPS) spectra of O1s core level spectra of (**a**) as-grown Ga_2_O_3_ films and Ga_2_O_3_ films annealed at (**b**) 700 °C, (**c**) 800 °C, and (**d**) 900 °C.

**Figure 4 sensors-20-06159-f004:**
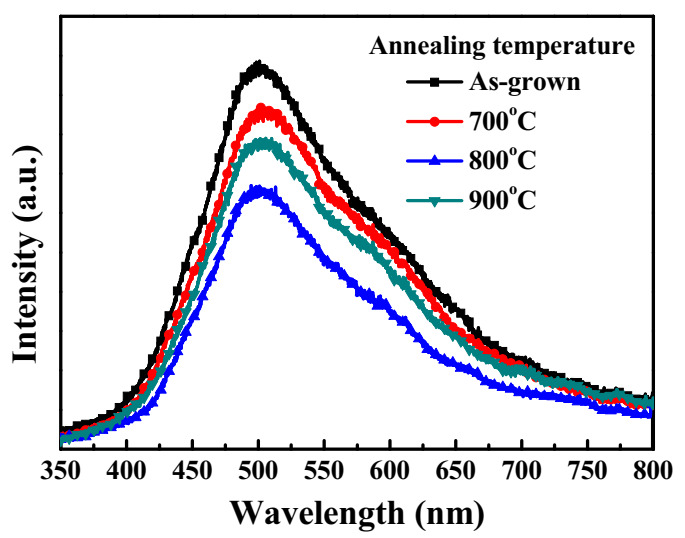
Room-temperature photoluminescence (PL) spectra of as-grown Ga_2_O_3_ films and Ga_2_O_3_ films annealed at various temperatures.

**Figure 5 sensors-20-06159-f005:**
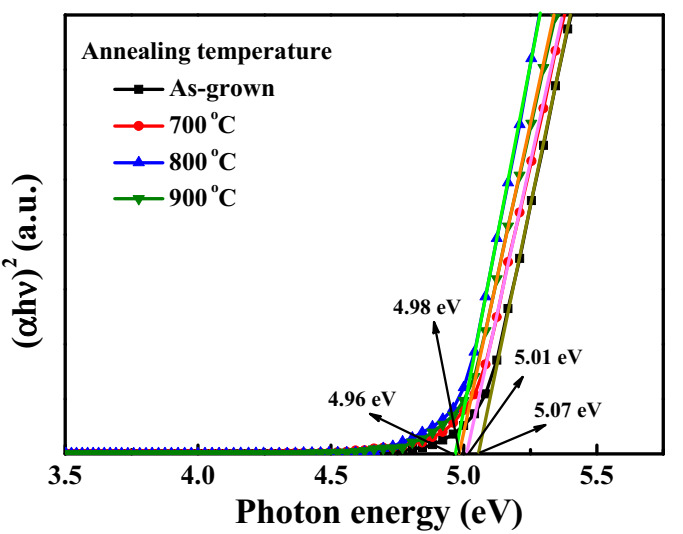
The (*αhυ*)^2^ versus *hυ* characteristics of as-grown Ga_2_O_3_ films and Ga_2_O_3_ films annealed at various temperatures.

**Figure 6 sensors-20-06159-f006:**
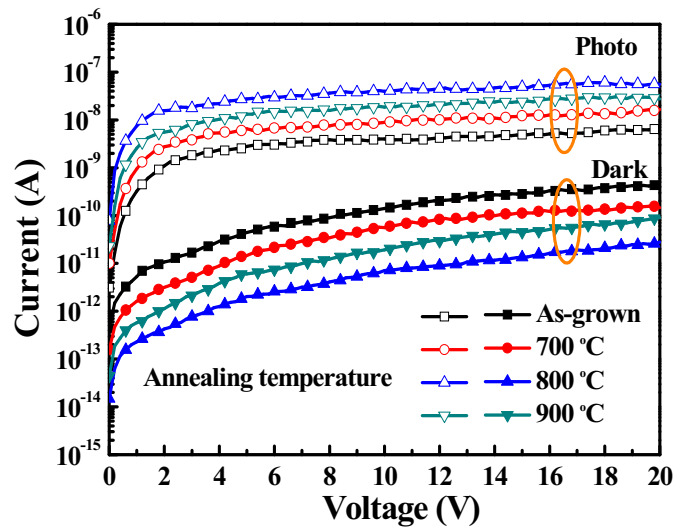
Dark current and photocurrent of various Ga_2_O_3_-based metal-semiconductor-metal ultraviolet C photodetectors (MSM UVC-PDs) as a function of bias voltage.

**Figure 7 sensors-20-06159-f007:**
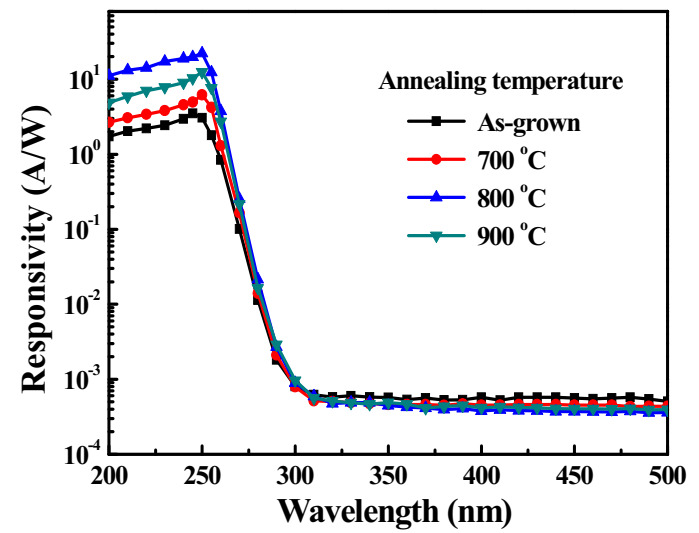
Photoresponsivity spectra of various Ga_2_O_3_-based MSM UVC-PDs.

**Figure 8 sensors-20-06159-f008:**
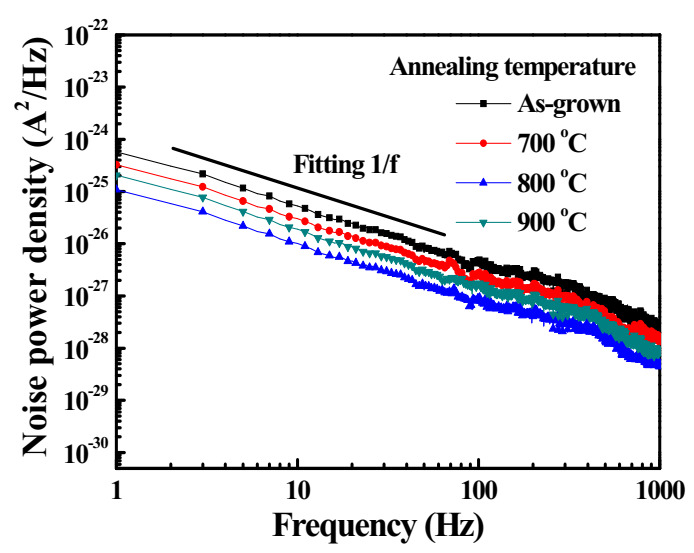
Noise power density of various Ga_2_O_3_-based MSM UVC-PDs as a function of frequency under a bias voltage of 5 V.

**Table 1 sensors-20-06159-t001:** Comparison of the performance of metal oxide-based MSM PDs.

Materials and Structure	Photoresponsivity (A/W)	UV/Visible Rejection Ratio	Detectivity (cmHz^1/2^W^−1^)	Ref
Ga_2_O_3_ thin films	22.19 (at 250 nm)	5.97 × 10^4^	8.74 × 10^12^	This work
Ga_2_O_3_ nanostructures	38.161 (at 365 nm)	–	8.39 × 10^9^	[29]
Ga_2_O_3_ thin films	17 (at 255 nm)	8.5 × 10^6^	7.00 × 10^12^	[30]
Ga_2_O_3_ thin films	0.893 (at 250 nm)	9.75 × 10^2^	–	[31]
Ga_2_O_3_ thin films	29.8 (at 254 nm)	9.4 × 10^3^	1.00 × 10^12^	[32]
α/β-Ga_2_O_3_ nanorods	0.00026 (at 254 nm)	2.7 × 10^3^	2.8 × 10^9^	[33]
TiO_2_ thin films	13.29 (at 365 nm)	–	4.91 × 10^13^	[34]
MgZnO thin films	0.14 (at 325 nm)	3.83 × 10^3^	4.42 × 10^12^	[35]
ZnO thin films	27 (at 365 nm)	–	8.5×10^13^	[36]

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
