# Peer review of "Investigation of Ga2O3-Based Deep Ultraviolet Photodetectors Using Plasma-Enhanced Atomic Layer Deposition System"

_sensors, 2020, doi:10.3390/s20216159_

Round 1
Reviewer 1 Report
The authors describe an article entitled “Investigation of Ga2O3-based deep ultraviolet photodetectors using plasma-enhanced atomic layer deposition system”. The topic of the manuscript is interesting, and the manuscript constitutes an interesting study concerning the development of deep-UV photodetectors.
The work is well written, and sufficient spectra and figures are included in the manuscript for comprehension and clarity. Convincing results have been obtained in this work. Therefore, I recommend accepting the article after inclusion of minor revision.
1) What about the device stability over time? This point should be commented.
2) During the measurements of the dark current and photocurrent of various Ga2O3-based MSM UVC-PDs (Figure 6), the voltage only range from 0 to 5V ? Why not to examine the photogenerated current at higher voltages ?
3) If the results obtained with these devices are interesting, a comparison with other oxide-based devices should be established. Indeed, no comparison with date reported in the literature is provided.
4) What about the excited state lifetime and the photoluminescence quantum yield? At present, only the photoluminescence spectrum has been recorded.
For all the above-mentioned reasons at present do not publish.
Reviewer 2 Report
This manuscript proposes an Investigation of Ga2O3-based deep ultraviolet photodetectors using plasma-enhanced atomic layer deposition system. Some experimental results are interesting. However, the overall quality of the manuscript is low due to the lack of scientific insights and there is a line of technical details that the reader can not find in the manuscript apparently necessary to understand the matter for example:
- The sensing mechanism is unclear. The authors must present a photo of the experimental setup.
- The manuscript has writing errors
- What is the repeatability of the measurements?
- Based on the obtained results, it is really not impressive, and most of the commercial product able to function better than this. Authors should highlight the novelty of the obtained results, particularly in the potential application.
- The author must show simulation about the principle operation of the system.
- Authors must improve all figures
- Authors should compare their results with those reported in the literature in a table
Round 2
Reviewer 2 Report
Accept